# Fast and Effective Removal of Congo Red by Doped ZnO Nanoparticles

**DOI:** 10.3390/nano13030566

**Published:** 2023-01-30

**Authors:** Biplob Kumar Pramanik, Nahar Singh, Rumbidzai Zizhou, Shadi Houshyar, Ivan Cole, Hong Yin

**Affiliations:** 1CSIR-National Physical Laboratory, Bhartiya Nirdeshak Dravya Division, New Delhi 110012, India; 2Academy of Scientific and Innovative Research (AcSIR), Ghaziabad 201002, India; 3School of Engineering, Royal Melbourne Institute of Technology, RMIT University, Melbourne, VIC 3000, Australia; 4Center for Materials Innovation and Future Fashion (CMIFF), School of Fashion and Textiles, Royal Melbourne Institute of Technology, RMIT University, Melbourne, VIC 3000, Australia

**Keywords:** nanoparticles, zinc oxide, wastewater treatment, dye removal, alginate membrane

## Abstract

ZnO nanoparticles (NPs) show remarkable efficiency in removing various contaminants from aqueous systems. Doping ZnO NPs with a second metal element can dramatically change the physicochemical properties of the pristine nanoparticles. However, there have been limited reports on the absorption of doped ZnO NPs, especially comparing the performance of ZnO NPs with different doping elements. Herein, ZnO NPs were doped with three transitional metals (Co, Fe, and Mn) at a nominal 2 wt.%. The particle surface had a higher dopant concentration than the interior for all NPs, implying the migration of the dopants to the surface. Because doping atoms inhibited grain growth, the doped ZnO NPs had a small particle size and a large surface area. The adsorption performance followed the order of Fe-doped < undoped < Mn-doped < Co-doped ZnO. Co-doped ZnO had an increased surface area and less tendency to agglomerate in an aqueous solution, showing the best adsorption performance. The adsorption of Congo red (CR) on Co-doped ZnO followed the pseudo-second-order model and the Langmuir isotherm. The adsorption process was spontaneous through monolayer chemisorption, and the maximum adsorption capacity was 230 mg/g. Finally, the Co-doped ZnO was successfully incorporated into an alginate membrane by electrospinning. The membrane demonstrated excellent adsorption performance and had great potential as an innovative and low-cost adsorbent (inexpensive raw materials and simple processing) for wastewater purification.

## 1. Introduction

Untreated wastewater generated by industries poses a threat to the ecosystem. Each year, the industrial sector releases around 300–400 Mt of waste into water bodies [1], which, if not properly treated before release, can have a negative impact on the environment and human health. Synthetic dyes from the industries, such as the textile and food industries, are among the primary sources of contaminants [2]. When dye-containing industrial wastewater is discharged without proper treatment, a number of issues arise. First, it lowers the penetration of sunlight in water bodies, reducing the photosynthesis process and hence jeopardizing the health of the aquatic ecosystem by lowering dissolved oxygen in water bodies [3]. Second, some dyes, such as Congo red (CR), are highly toxic [4] and persist as environmental contaminants in food chains, resulting in unintended biomagnification [5]. Conventional wastewater treatment methods may be insufficient to meet the increasingly stringent legislation requirements [1]. Some novel methods have been reported to recover nutrients from wastewater, such as phosphorous. However, only a very limited number of contaminants are worth recovering economically, and elimination is the most simple and straightforward way to ensure safe discharge into the natural environment [6]. Therefore, cost-effective and efficient wastewater treatment technologies are necessary.

Over the last few decades, many advanced technologies, such as nanofiltration [7], electrodialysis [8], electrocoagulation [9], bioremediation [5], ozonation [10], Fenton reaction [11], adsorption [12], and ion exchange [3], have been employed to eliminate dyes from water. Although these methods are effective, they have a number of limitations, including the consumption of large quantities of chemicals and energy, high processing costs, and the production of large quantities of secondary waste [13]. Additionally, the majority of these advanced techniques are ineffective for the removal of ultra-low levels of dyes from water. Therefore, additional efforts are still required to develop efficient wastewater treatment methods. Adsorption is one of the ancient and most significant methods for removing toxic contaminants from wastewater. It is considered one of the most efficient methods due to its versatility, simplicity of design, adsorbent reuse potential, low cost, and eco-friendliness and has been extensively studied and widely used in full application [14,15]. Nanomaterials-based adsorbents are potential adsorbents because of their singular physical and chemical features, including a moderate synthesis temperature [16], high surface area, high adsorption capacity, reasonable recyclability, and chemical and thermal stability. Various nano-adsorbents, such as charred biomass [17], carbon dots [18], CNTs [19], zeolites [20], metal oxides [21], activated carbon [22], organic frameworks [23], and porous materials [24], have been reported.

Zinc oxide (ZnO) nanomaterials feature unique catalytic, UV-absorbing, electrical, optoelectronic, and photocatalytic characteristics [25]. ZnO nanoparticles (NPs) also show remarkable efficiency in removing heavy metal ions, azo dyes, methyl orange, and amaranth from aqueous systems [26]. Doping ZnO NPs with a secondary metal element can dramatically change the physicochemical properties of the pristine ZnO NPs. It is expected that appropriately doped ZnO NPs can have a better adsorption performance than undoped ZnO NPs [27,28]. According to our best knowledge, there has been no reported literature on the doping effect on the absorption performance of ZnO NPs.

In this study, ZnO NPs were doped with three transitional metals (Co, Fe, and Mn) at a nominal 2 wt.%, aiming to investigate the efficiency of differently doped ZnO nanomaterials as potential absorbents for effectively eliminating the pollutant dye, CR, from wastewater. This was the first laboratory study to compare the efficiency of Fe-, Mn-, or Co-doped ZnO nanomaterials in the remediation of CR from wastewater. Our results demonstrated that Co-doped ZnO had the best adsorption performance. Therefore, in order to derive the adsorption kinetic and isotherm models for Co-doped ZnO NPs, adsorption parameters, including pH, adsorbent dosage, contact time, and starting dye concentration, were tuned. Finally, the Co-doped ZnO NPs were successfully incorporated into an electrospun alginate membrane. To enhance sorption capacity, functional substituents on alginate chains (carboxylate and hydroxyl groups) served as binding sites for target analytes.

## 2. Materials and Methods

### 2.1. Synthesis of ZnO and Doped ZnO NPs

Undoped and Fe-, Mn-, and Co-doped ZnO NPs were synthesized using a simple sol–gel method. Compared with commercially available milled ZnO, NPs prepared using the sol–gel method had more active surface groups for the adsorption of pollutants. It was also easy to incorporate doping elements using the wet-chemistry method [16].

All precursors were purchased from Sigma-Aldrich (Castle Hill, NSW, Australia). For undoped ZnO NPs, Zn(CH_3_COO)_2_·2H_2_O (5.48 g) was dissolved in 50 mL of methanol to obtain a solution of 0.5 M. Then, 1 mL of NaOH solution (1 mol/L) was added to the mixture dropwise, with continuous stirring at 80 °C for 2 h, and white powders were precipitated during stirring. The obtained powders were centrifuged and washed many times with deionized (DI) water until the solution was neutral. The powders were dried at 120 °C and subsequently annealed at 350 °C for 2 h. For the synthesis of doped ZnO NPs, precursors containing iron (II) acetate (0.102 g), manganese (II) acetate (0.104 g), and cobalt (II) acetate (0.099 g) were used to prepare mixed solutions in methanol to a nominal 2 wt.% in the nanomaterials. The actual doping concentration in the final powder was determined after synthesis.

### 2.2. Characterization of the Synthesized ZnO and Doped ZnO NPs

Inductively coupled plasma atomic emission spectroscopy (ICP-AES, Varian Vista AX Simultaneous Axial) was used to determine the precise concentration of doping in the NPs following digestion. X-ray diffraction (XRD) analysis was performed on a Bruker D8 Advance diffractometer to determine the presence of crystalline phases. The immediate atomic environment was studied using extended X-ray absorption fine structure (EXAFS). Images captured by a transmission electron microscope (TEM, JEOL, 100CX-II, Tokyo, Japan) were analyzed to determine the particle morphologies. Approximately 10 mg of particles were suspended in 5 mL of deionized water. A drop of the dispersion was deposited on a carbon-coated copper grid and dried overnight after being ultrasonically treated for 30 min. Approximately a hundred particles’ diameters were measured to evaluate the particle size and distribution. X-ray photoelectron spectroscopy (XPS; ESCA LAB 220i-XL Thermo VG Scientific, West Sussex, UK) was used to examine the surface chemistries of the samples. CasaXPS software (2.3.25, Casa Software Ltd., Teignmouth, UK) was utilized to process XPS data files. A Malvern Nano Z Zetasizer was used to measure the zeta potentials and hydrodynamic diameters of the ZnO NPs in deionized water. For good dispersion, 10 mg of ZnO NPs were placed in a cuvette with 3 mL of deionized water, sonicated for 10 s, and then shaken by hand. Each sample was measured five times to receive an average of its zeta potential and hydrodynamic diameter.

### 2.3. Adsorption

First, 500 mg of CR dye powder was dissolved in 1000 mL of distilled water with constant stirring for 10 h to create a 500 mg/L CR stock solution. In order to slow the stock solution’s deterioration, it was kept in the dark.

Several adsorption parameters, including pH, adsorbent (Co-doped ZnO) dose, contact time, and adsorbate (CR dye) dose, were optimized by performing batch adsorption tests. A known amount of Co-doped ZnO was added to a defined volume of CR dye solution to optimize adsorption parameters. The dye solutions with Co-doped ZnO were stirred for 12 min in the dark at room temperature, followed by centrifugation at about 8000 rpm for 5 min to separate the Co-doped ZnO from the solution. A UV–Vis spectrophotometer was then used to measure the concentration of the supernatant.

To establish the maximum dye removal by Co-doped ZnO in a batch adsorption experiment, we used Equations (1) and (2) to calculate the amount of CR removed and the adsorption capacity of Co-doped ZnO.

The percentage removal of CR dye was calculated by
(1)$$\%R=\frac{\left(C_{i}-C_{f}\right)\times 100}{C_{i}}$$

Adsorption capacity was calculated by
(2)$$q_{e}=\frac{\left(C_{i}-C_{f}\right)V}{W}$$
where C_i_ and C_f_ are the initial and final dye concentrations (mg/L), V is the volume in milliliters of CR dye solution, W is the weight in grams of the adsorbent, and q_e_ is the adsorption capacity in milligrams per gram.

Using Co-doped ZnO as an example, the schematic representation of CR dye removal is shown in Figure 1.

### 2.4. Electrospun Film

To fabricate the electrospun film, a 0.25 wt.% Co-doped ZnO NP suspension in deionized water was ultrasonicated for 3 h at room temperature. Subsequently, polyethylene oxide (PEO) and sodium alginate (SA) were then added at a ratio of 70:30 to make a 3.5 wt.% aqueous solution. To increase the spinnability of the solution, Triton X100 was then added at a concentration of 1 wt.%. Electrospinning was conducted at a voltage of 10 kV, a flow rate of 0.2 mL/h, and a distance of 15 cm. The electrospun membrane was collected on a flat aluminum plate. After electrospinning, the membrane was peeled off the collector and immersed in absolute ethanol at 70 °C for 15 h to remove the PEO. To enable crosslinking of the alginate, the mat was immersed in a solution containing 3 wt.% of Ca^2+^ for 30 s. The electrospun mat was then left to dry at 50 °C for 24 h.

## 3. Results and Discussion

### 3.1. Characterization of Doped ZnO

Quantitative analysis for Fe-doped, Mn-doped, and Co-doped ZnO is presented in Table 1 to compare the nominal dopant concentration (used for synthesis), actual dopant concentration (measured using ICP-MS), and dopant concentration detected on the surface (measured by XPS). The real doping amount in ZnO NPs analyzed by ICP-MS was 2.1 wt.% for Fe, 0.8 wt.% for Mn, and 1.1 wt.% for Co. Because the precursors containing Fe, Mn, and Co were utilized for the synthesis to obtain a nominal 2 wt.% in all doped ZnO NPs, the results indicated that Fe was easier to incorporate into the ZnO lattice. The particle surface analyzed by XPS was evident in the presence of doping elements. A higher dopant concentration was found on the particle surface for Fe, Co, and Mn, implying the migration of the dopants to the surface of the NPs.

The Fourier transform amplitude vs. the radial distance of the Zn K-edge is shown in Figure 1 as a result of the local atomic environment being examined with EXAFS. All types of ZnO NPs displayed two notable peaks at 1.5 and 2.9 Å, corresponding to the Zn–O and Zn–Zn shells, respectively. All ZnO NPs had the same Zn–O shell intensity (R = 1.5 Å). Co-doped ZnO had a very similar spectrum to that of ZnO, indicating that Co ions substituted for Zn^2+^ sites in the lattice without forming a secondary phase. The slight shift to a lower distance for Co-doped ZnO in the second peak indicated an interaction between Zn–Co instead of Zn–Zn. The intensity of the Fourier transform magnitude (y-axis) was significantly lower for the Mn-doped and Fe-doped ZnO than that for the undoped ZnO in the Zn–Zn shells, suggesting a lower coordination number in both Fe- and Mn-doped particles. It implied that the local environment of Zn changed. The spectra of Mn-doped ZnO and Fe-doped ZnO were very similar, with the exception of a larger Fourier transform amplitude in the Zn–Zn shell for the latter, which agreed well with the higher Fe concentration in Fe-doped ZnO NPs.

Both doped and undoped ZnO NPs were analyzed for their morphology using TEM. Figure 2 displays the resulting TEM images. Particles of doped and undoped ZnO were nearly identical in shape and size distribution. However, undoped ZnO NPs had larger sizes than the doped ones. By taking the average of the diameters of 100 particles, undoped ZnO NPs had a primary particle size of 25.8 nm, while Fe-doped ZnO NPs were 12.2 nm, Mn-doped ZnO NPs were 16.5 nm, and Co-doped ZnO NPs were 17.0 nm. Sahu et al., reported that doping atoms could inhibit grain growth, leading to smaller particle sizes of the doped NPs [29]. It was not a surprise that Fe-doped ZnO had the smallest particles because it had the highest amount of doping (2.1 wt%). Table 2 also shows that doping significantly increases the surface area, which is expected to enhance the adsorption performance.

For the purpose of investigating surface charge in aqueous media, the zeta potentials determined in deionized water (pH = 6.0) are listed in Table 2. According to the report that the isoelectric point of ZnO was between 8 and 9.33, the zeta potentials of undoped, Fe-doped, Mn-doped, and Co-doped ZnO NPs were 16.1, 12.2, 10.6, and 12.1 mV, respectively. The primary particle size acquired by TEM is typically between 10 and 30 nm, but the hydrodynamic diameters measured in water ranged from 500 to 700 nm, indicating the presence of a large number of agglomerations.

### 3.2. Adsorption of Three Different Doped ZnO

Figure 3 shows the UV-Vis spectra of 100 ppm of CR solution with and without ZnO NPs. The pure CR solution demonstrated two distinct peaks at 350 nm and 500 nm, respectively. The red color faded after adding ZnO NPs. The spectrum change was due to the degradation of CR, as ZnO is a highly efficient photocatalyst and has been reported to degrade various dyes under the exposure to UV or visible light [30,31]. The adsorption performance followed the order of Fe-doped < undoped < Mn-doped < Co-doped ZnO. Doping increased the surface area of ZnO NPs. However, Fe-doped NPs had lower adsorption due to their tendency to agglomerate, demonstrated by the hydrodynamic diameter and the TEM images. Co-doped ZnO had an increased surface area and smaller hydrodynamic diameter, showing the best adsorption performance. In the following section, detailed adsorption studies were performed on Co-doped ZnO, including the optimization of adsorption parameters, kinetic and thermodynamic studies, and isotherm models.

### 3.3. Adsorption Study of Co-Doped ZnO NPs

#### 3.3.1. pH Optimization

The percentage removal of CR by Co-doped ZnO in 25 mL of CR solution (50 mg/L) was investigated by adjusting the solution pH from 2 to 10. Each solution was added with 0.03 g of ZnO. As shown in Figure 4a, the removal effectiveness of CR by Co-doped ZnO was at the maximum at low pH and decreased with the increase in pH. At a high pH, negatively charged OH- ions on the surface of Co-doped ZnO repelled the anionic CR dye molecule. Similarly, there was an electrostatic attraction between Co-doped ZnO and CR dye molecules at lower pH levels. The zeta potential measurements of Co-doped ZnO showed that its zero-point charge (p_zpc_) was between 8 and 9. This implies that Co-doped ZnO has a positive surface charge at lower pH (<p_zpc_) levels and a negative surface charge at higher pH (>p_zpc_) levels. Based on adsorption data, it was inferred that Co-doped ZnO NPs removed CR dye over a broad pH range, and the maximum removal of 99.8% was obtained for the 100 mg/L CR dye solution.

#### 3.3.2. Adsorbent Dosage Optimization

The removal percentage of CR by Co-doped ZnO was investigated by increasing the adsorbent dose from 0.01 to 0.035 g for the same dye volume and concentration. The results are depicted in Figure 4b, demonstrating that the CR removal percentage increased with the increasing adsorbent dose and reached a plateau. A similar trend was reported by Sneha et al. [32] for the removal of CR dye using zinc peroxide. This can be explained by the fact that increasing the amount of Co-doped ZnO adsorbent increases the number of active free sites for adsorption. The highest percentage of CR elimination (99.8%) was achieved when 0.03 g of Co-doped ZnO was used.

#### 3.3.3. Contact Time Optimization

As illustrated in Figure 4c, the contact duration was adjusted from 1 to 12 min at the optimal adsorbent dose for the same dye volume and concentration. The percentage of CR removed by Co-doped ZnO increased as the contact duration increased. This was because the additional time allowed the CR dye to interact with Co-doped ZnO for a longer time. The removal rate peaked at 10 min, after which it remained constant. Therefore, Co-doped ZnO and CR dye need to come into contact for 10 min to achieve maximum adsorption.

#### 3.3.4. Initial Dye Concentration Optimization

The concentration of CR dye was tuned by evaluating the removal percentage of CR at the optimized adsorbent dose and time for varied starting concentrations, which ranged from 50 mg/L to 250 mg/L. As shown in Figure 4d, the removal percentage of CR dye decreased as the dye concentration increased. This drop was caused by the saturation of Co-doped ZnO’s active sites, following the adsorption of a specific concentration of CR dye. The adsorption capacity of Co-doped ZnO was determined, using Equation (2), to be 230 mg/g for a 250 mg/L concentration of CR dye.

#### 3.3.5. Kinetic Study

The factors influencing the adsorption rate can be learned from studying the adsorption kinetics. The rate-controlling step in an adsorption experiment can be determined with the help of adsorption kinetic models, such as the pseudo-first-order and pseudo-second order models. The linear relationships for the pseudo-first-order model and the pseudo-second-order model are shown in Equations (3) and (4), below.
(3)$$\ln\left(q_{e}-q_{t}\right)=\ln q_{e}-K_{1}t$$
(4)$$\frac{t}{q_{t}}=\frac{1}{K_{2}q^{2}{}_{e}}+\frac{t}{q_{e}}$$
where q_t_ is the adsorbed dye at time t (mg/L), q_e_ is the adsorbed dye at equilibrium, and K_1_ and K_2_ are the pseudo-first-order and pseudo-second-order rate constants, respectively.

The kinetic data are shown in Figure 5a,b. Comparing the R^2^ values of the two models, the pseudo-second-order model better fits the data, indicating that the process leaned more toward chemisorption. Additionally, the comparison of calculated and theoretical q_e_ values validated the superiority of the pseudo-second-order model over the pseudo-first order-model. In addition, the intraparticle diffusion model was used to analyze the adsorption process, as shown in Equation (5) [33].
(5)$$q_{t}=C+K_{3}t^{1/2}$$
where K_3_ is the rate constant (mg g^−1^ h^−1/2^) and C is a constant related to the boundary thickness.

The result in Figure 5c illustrates a three-step adsorption process involving bulk diffusion to the surface of the oxide particle, surface adsorption, and porous diffusion. The negatively charged CR dye molecule diffused towards the positively charged Co-doped ZnO (p_zpc_ = 8–9) and was then adsorbed on the surface of the doped ZnO adsorbent. Nevertheless, the extension of the graph did not pass the origin, showing that this model was not the only plausible model for explaining the current adsorption process.

Table 3 summarizes the values of the critical constants and parameters derived for the three models.

#### 3.3.6. Isotherm Study

The Langmuir, Freundlich, Toth, Redlich–Peterson, and Temkin models could be used to describe the adsorption isotherms. Linear versions of the Langmuir, Freundlich, and Temkin models were employed here. Each model’s representative linear equations are shown in Equations (6)–(8),
(6)$$\frac{C_{f}}{q_{e}}=\frac{1}{K_{L}Q_{m}}+\frac{C_{f}}{Q_{m}}$$
(7)$$\ln q_{e}=\ln K_{f}+\frac{1}{n}\ln C_{f}$$
(8)$$q_{e}={\mathrm{BlnK}}_{T}+B\ln C_{f}$$
where C_e_ and q_m_, respectively, stand for the equilibrium adsorbate concentration and the maximum adsorption capacity. The sorption energy, maximum absorption capacity, and sorption heat are represented by K_L_, K_f_, and K_T_, respectively, for the Langmuir, Freundlich, and Temkin constants.

As demonstrated by adsorption isotherms, an adsorbent’s maximum adsorption capacity is a function of dye concentration. Figure 6 depicts the CR adsorption isotherm for Co-doped ZnO. The as-synthesized Co-doped ZnO had a strong adsorption capacity for CR at various starting concentrations, with a maximum estimated adsorption capacity of 230 mg/g.

The adsorption mechanism and the homogeneity or heterogeneity of the adsorbent surface can be determined with the aid of adsorption isotherms, which define the adsorbent’s interaction with the adsorbate and connect the experimental results to the adsorption model. In this study, we analyzed experimental adsorption data using the Langmuir, Freundlich, and Temkin models. A study of the correlation coefficient revealed that the Langmuir model provided the best fit to the data (R^2^). This exemplifies how the Co-doped ZnO surface formed a monolayer of CR molecules and was uniform throughout. Co-doped ZnO’s exceptional adsorption performance can be attributed in large part to the presence of functional groups near its surface. Adsorption isotherm values are shown in Table 4 for a variety of models; the Temkin model accounted for a lot of electrostatic interactions. Furthermore, CR adsorbed preferentially onto Co-doped ZnO, as shown by the number 1/n < 1 in the Freundlich model. The interactions of CR dye with ZnO are depicted in Figure 1.

#### 3.3.7. Thermodynamic Study

The influence of temperature was tested for 10 min at temperatures ranging from 10 to 70 °C using a 50 mg/L dye solution and 0.03 g of adsorbent. The equilibrium constant K_c_ was computed using the values of C_o_ and C_f_ according to Equation (9).
(9)$$K_{c}=\frac{C_{o}}{C_{f}}$$
where C_o_ represents the equilibrium adsorption in mg/L, and C_f_ stands for the CR dye concentration at equilibrium in mg/L.

Equation (10)’s slope and intercept were used to calculate the values of other thermodynamic parameters, such as ∆H and ∆S, while Equation (11)’s free energy (∆G) values at various temperatures were calculated.
(10)$$\mathrm{lnK}{}^{\circ}=\frac{\Delta S{}^{\circ}}{R}-\frac{\Delta H{}^{\circ}}{\mathrm{RT}}$$
(11)ΔG°=−RTlnK°
where T represents the temperature in K, and R represents the gas constant in kJ/molK.

The graph obtained and the experimental data for thermodynamic parameters are shown in Figure 7 and Table 5, respectively. It was evident that both ∆H and ∆S were negative. The negative value of ∆H for CR removal by Co-doped ZnO confirmed that the adsorption process was exothermic, while the negative value of ∆S indicated that the entropy decreased upon the adsorption of CR dye on Co-doped ZnO. The negative ∆G° values suggested that the current adsorption process was thermodynamically favorable.

#### 3.3.8. Comparison with the Adsorbents Reported for CR Removal in Literature

Table 6 compares the maximum adsorption capabilities of the adsorbents reported for CR dye removal in literature and the ∆H and ∆G° of their adsorption processes. Clearly, the Co-doped ZnO used in this study had a higher adsorption capacity, making it a potential material for removing anion dyes from aqueous solutions. The existence of electrostatic interaction between the metal oxides and the dye prompted the adsorption capability. The most negative value of ∆H confirmed that the adsorption process was exothermic, and the most negative ∆G° value suggested that the adsorption process was most thermodynamically favorable, compared to the other absorbents.

#### 3.3.9. Stability Test

In order to make an adsorbent cost-effective for practical use, it is crucial for it to have high regeneration capabilities. In this research, regeneration experiments were conducted using sonication for half an hour in ethanol to treat CR-adsorbed, Co-doped ZnO NPs. After the desorption process, the regenerated Co-doped ZnO NPs were recycled and used for the adsorption of CR dye again. The desorption–adsorption cycle was repeated five times. As shown in Figure 8, the Co-doped ZnO NPs retained more than 82% of their original adsorption capacity, even after five cycles of desorption–adsorption (Figure 8a). This demonstrated that Co-doped ZnO NPs had high levels of reusability for the treatment of dye pollutants. The FTIR spectra suggested that the -OH groups were significantly reduced after recycling (broad peak at 3450 cm^−1^), confirming the interaction of CR with Co-doped ZnO. Other than the -OH groups, the chemical states were not changed before and after five times of adsorption (Figure 8b). XRD patterns also suggested a reduced amount of zinc hydroxides (some minor changes at 2θ = 27.9 and 31.3°). The other main diffraction peak had no significant differences before and after recycling, suggesting that the crystallographic structure remained stable (Figure 8c).

### 3.4. Incorporation of Co-Doped ZnO Electrospun Film

Figure 9 confirms that the Co-doped ZnO was successfully incorporated into a polymeric membrane. The particles showed agglomerations in the matrix. In the FTIR spectra, the sodium alginate membrane showed carboxylic (strong peak at 1720 cm^−1^) and hydroxyl groups (broad peak at 3245 cm^−1^). After introducing Co-doped ZnO NPs into the membrane, these peaks were weakened, suggesting the interaction between the sodium alginate membrane and the NPs.

Figure 10 compares the uptake effect of Co-doped ZnO NPs and Co-doped ZnO alginate film. It is clear that when Co-doped ZnO was incorporated into an alginate membrane, the removal efficiency increased. This increase was due to the combined effect of Co-doped ZnO and alginate. The typical texture of the electrospun alginate membrane can enhance contact area, prompting molecule diffusion. In addition, the carboxylate and hydroxyl groups on the alginate chains can serve as binding sites for target analytes, hence enhancing the sorption capacity.

## 4. Conclusions

ZnO NPs were doped with three transitional metals (Co, Fe, and Mn) at a nominal 2 wt.%. Due to the increased specific surface area and reduced agglomeration, Co-doped ZnO NPs demonstrated the highest removal rate for CR dye from water. Among all the isothermic models tested, the Langmuir isotherm best fit experimental data. The maximum CR dye adsorption capacity of Co-doped ZnO NPs was 230 mg/g. The pseudo-second-order model adequately described the rate mechanism of dye adsorption. According to thermodynamic research, the adsorption of CR dye onto Co-doped ZnO NPs is spontaneous through chemisorption. Our finding provided a simple way to enhance the absorption performance of ZnO NPs by introducing a small amount of Co precursor in the sol–gel process. Additionally, the Co-doped ZnO in alginate membrane also showed excellent CR removal efficiency, confirming the potential of electrospun membranes with Co-doped ZnO NPs as an innovative and low-cost adsorbent for wastewater purification.

## Data Availability

Not applicable.

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
