# Peer review of "Fast and Effective Removal of Congo Red by Doped ZnO Nanoparticles"

_nanomaterials, 2023, doi:10.3390/nano13030566_

Round 1

Reviewer 1 Report

This paper reported the preparation of doped ZnO and their practical use as adsorbents for dye removal. The topic of the work was significant with regard to the advancement of environmental purifications. The manuscript can be considered for publication after the following comments are addressed.

Comments:

(1) The figure quality in terms of resolution and esthetics should be improved.

(2) Labels should be marked for the four panels of Fig 2. Some scale bars were missed.

(3) The adsorption capacity of the samples should be normalized to the surface area, with which factors other than surface area contributing to the best adsorption capacity of Co-doped ZnO may be revealed.

(4) The comparison in Table 4 should include thermodynamics data, i.e. enthalpy change and free energy change. Otherwise, it is difficult to appreciate the merits of the proposed doped ZnO.

(5) How about the stability of the samples? Recycling experiments are needed. The chemical states and crystallographic structure of the samples upon repeated use should also be examined.

(6) Photocatalysis stands for another powerful tool of removing dye molecules. Recent review article summarizing the advancement of photocatalysis technology on dye removal should be introduced: Catalysts 2019, vol.9, pp.430.

Reviewer 2 Report

Dear author(s),

these are some inspiring insights in this manuscript and I tend to support its publication. However, some comments needs to be addressed (to improve its communication in particular) before I can give my green light to its publication:

Abstract:

1/ you argue about "low cost", however, this argument is not sufficiently proved in the manuscript; remove such claim or (better) provide breakdown of the manufacturing process

2/ the industrial significance of this work is poorly communicated, better explain how will our audience of readers benefit from these revelations

Introduction:

3/ remove typos such as "electrodialysis[7]" = electrodialysis [7] etc.

4/ deeper review the latest trends in wastewater management such as nutrient recovery, refer to paper "Recovering phosphorous from biogas fermentation residues indicates promising economic results"

5/ better explain to our readers whether nanoparticles are commonly used in wastewater treatment plants (what are the advantages and disadvantages?)

6/ it should be noted that pricing of metals (rare metals in particular) affects the production cost of "NPs", refer to paper "Predicted Future Development of Imperfect Complementary Goods – Copper and Zinc Until 2030"

7/ do not ignore the (economic) reality, comment on the costs associated with the use of wastewater treatment via all the reviewed technologies

8/ kindly note that charred biomass can be also used as a sorbent/carrier, refer to paper "Techno-economic analysis reveals the untapped potential of wood biochar"

9/ the research hypothesis could be formulated more straightforward

Materials and methods:

10/ make sure that anybody who reads this chapter is capable to replicate all your procedures (provide more detail to each step)

11/ provide analysis on cost breakdown if you want to argue about cost competitiveness

Results and discussion:

12/ show more criticism to own work, what are the limitations of the methods used?

13/ deeper compare your production methods with other latest techniques reported in the literature, refer to paper "Silica nanoparticles from coir pith synthesized by acidic sol-gel method improve germination economics"

14/ make all the charts look more uniform/similar

15/ it should be noted that the use of iron in water can result in some environmental issues, refer to paper "Economic impacts of soil fertility degradation by traces of iron from drinking water treatment"

16/ Table 4: try to comment on the cost

17/ do not neglect the aspects related to the transfer of technology to the commercial sphere, comment on the economic aspects related to mass production (refer to paper "Artificial Intelligence Data-driven Internet of Things Systems, Real-Time Advanced Analytics, and Cyber-Physical Production Networks in Sustainable Smart Manufacturing")

18/ "ml" = mL

Conclusions:

19/ do not repeat your methods and results again and again (this is not an Abstract)

20/ provide deeper synthesis of your results, higher level of generalization is advisable

Reviewer 3 Report

I have found this article interesting and well corresponding to the journal. Nevertheless, some improvement is still needed, which can be managed during the revision.
1. Would it be necessary to use nanoparticles? Why not use milled crystalline ZnO? It is available on the market, can be doped with 3d elements, and milled to an average particle size below one micron. A comment on topic is required.
Fig.3:. Adsorption  performance by measuring the absorption is shown. Why  spectrum is changed when different dopings are applied ? It has to be commented on.
Fig. 9: The curves for Co doped samples have to be amplified. Use multiplying, like x10. It seems that absorption spectra are changed as well. It should be commented on.

Round 2

Reviewer 1 Report

The revised manuscript is now in a good shape for publication.